# Impact of the Stroma on the Biological Characteristics of the Parenchyma in Oral Squamous Cell Carcinoma

**DOI:** 10.3390/ijms21207714

**Published:** 2020-10-18

**Authors:** Kiyofumi Takabatake, Hotaka Kawai, Haruka Omori, Shan Qiusheng, May Wathone Oo, Shintaro Sukegawa, Keisuke Nakano, Hidetsugu Tsujigiwa, Hitoshi Nagatsuka

**Affiliations:** 1Department of Oral Pathology and Medicine Graduate School of Medicine, Dentistry and Pharmaceutical Science, Okayama University, Okayama 7008525, Japan; gmd422094@s.okayama-u.ac.jp (K.T.); p4628fuz@s.okayama-u.ac.jp (H.O.); hrbmushanqiusheng@163.com (S.Q.); p1qq7mbu@s.okayama-u.ac.jp (M.W.O.); gouwan19@gmail.com (S.S.); keisuke1@okayama-u.ac.jp (K.N.); tsuji@dls.ous.ac.jp (H.T.); 2Department of Oral and Maxillofacial Surgery, Kagawa Prefectural Central Hospital, Kagawa 7600065, Japan; 3Department of Life Science, Faculty of Science, Okayama University of Science, Okayama 7000005, Japan

**Keywords:** tumor stroma, tumor parenchyma, tumor microenvironment, biological characteristics

## Abstract

Solid tumors consist of the tumor parenchyma and stroma. The standard concept of oncology is that the tumor parenchyma regulates the tumor stroma and promotes tumor progression, and that the tumor parenchyma represents the tumor itself and defines the biological characteristics of the tumor tissue. Thus, the tumor stroma plays a pivotal role in assisting tumor parenchymal growth and invasiveness and is regarded as a supporter of the tumor parenchyma. The tumor parenchyma and stroma interact with each other. However, the influence of the stroma on the parenchyma is not clear. Therefore, in this study, we investigated the effect of the stroma on the parenchyma in oral squamous cell carcinoma (OSCC). We isolated tumor stroma from two types of OSCCs with different invasiveness (endophytic type OSCC (ED-st) and exophytic type OSCC (EX-st)) and examined the effect of the stroma on the parenchyma in terms of proliferation, invasion, and morphology by co-culturing and co-transplanting the OSCC cell line (HSC-2) with the two types of stroma. Both types of stroma were partially positive for alpha-smooth muscle actin. The tumor stroma increased the proliferation and invasion of tumor cells and altered the morphology of tumor cells in vitro and in vivo. ED-st exerted a greater effect on the tumor parenchyma in proliferation and invasion than EX-st. Morphological analysis showed that ED-st changed the morphology of HSC-2 cells to the invasive type of OSCC, and EX-st altered the morphology of HSC-2 cells to verrucous OSCC. This study suggests that the tumor stroma influences the biological characteristics of the parenchyma and that the origin of the stroma is strongly associated with the biological characteristics of the tumor.

## 1. Introduction

Solid tumors consist of the parenchyma and stroma. The parenchyma–stroma complex forms the tumor microenvironment, which participates in the regulation of tumor progression, invasion, and metastasis [1,2]. In the conventional concept of oncology, the tumor parenchyma represents the tumor itself and defines the characteristics of the tumor tissue, whereas the tumor stroma plays a pivotal role in modulating tumor parenchymal growth and invasiveness, and is regarded as a supporter of the tumor parenchyma. However, in the past ten years or so, the tumor stroma has attracted attention because it includes structurally and functionally essential elements that participate in the regulation of solid tumor progression [3,4,5,6]. Cancer-associated fibroblasts (CAFs) in particular play key roles in tumor proliferation and invasion [7,8,9]. In addition, the tumor stroma also includes tumor-associated macrophages, endothelial cells, and T cells that contribute to tumor invasion and metastasis [10,11,12]. Thus, the tumor tissue is no longer regarded as a bulk of malignant cancer cells, but rather as a complex tumor microenvironment consisting of subpopulations of cells corrupted by tumor cells that form a self-sufficient biological structure.

Various subtypes of oral squamous cell carcinoma (OSCC) such as invasive carcinoma and verrucous carcinoma are described in the World Health Organization Classification of Head and Neck Tumors 4th Ed [13]. In addition, two macroscopic subtypes have also been identified based on the clinical invasion pattern: the exophytic type (EX-type) and the endophytic type (ED-type) [14,15]. These subtypes have important differences in prognosis due to differences in invasive ability. ED-type OSCC has the ability to invade and occasionally metastasize. On the other hand, EX-type OSCC such as verrucous OSCC shows outward growth, does not invade the subepithelial connective tissue, does not metastasize, and thus has relatively good prognosis. OSCC has different invasive, proliferative, and growth abilities depending on the macroscopic subtypes. The differences between the macroscopic subtypes are defined by cancer parenchyma properties such as invasion ability [16,17,18]. However, no studies have examined how the differences in subtypes affect the tumor stroma.

Although tumorigenesis is caused by accumulation of mutations, clinical tumorigenesis that cannot be explained by this mechanism occurs. For example, even if the epithelial tissue including the tumor and normal epithelium is removed as a precancerous lesion or a cancer lesion localized in the superficial region, recurrence may be observed even though no lesion is present under the epidermis. Some recurrences of OSCC arise from the skin flap after reconstruction following surgery for oral cancer [19,20]. Therefore, we hypothesized that the tumor stroma was affecting the characteristics of the tumor parenchyma. Generally, although the interaction between the tumor parenchyma and tumor stroma is thought to play important roles, only the parenchyma has been reported to control the stroma, and no report has described if or how the stroma regulates the parenchyma.

In this study, to investigate how the cancer stroma contributes to the properties of the cancer parenchyma, stromal cells isolated from OSCC EX-type and ED-type were co-cultured with the OSCC cell line (HSC-2) in vitro and in vivo, and we analyzed how the biological properties of the tumor parenchyma changed.

## 2. Results

### 2.1. Properties of Various Types of Stromal Cells

Normal dermoid fibroblasts (HDFs), ED-st (derived from ED-type OSCC stroma) and EX-st (derived from EX-type OSCC stroma) showed similar spindle-shaped morphology. HDFs were vimentin positive and α-smooth muscle actin (SMA) negative as seen with fluorescent immunohistochemical staining. On the other hand, both EX-st and ED-st were vimentin positive and partially α-SMA positive (Figure 1a). In addition, EX-st, ED-st and HDF were mouse anti-pan Cytokeratin (AE1/3 negative). Therefore, the stromal cells did not contaminate tumor cells (Figure 1a).

The proliferative activity of stromal cells and HDFs was compared with the MTS assay. HDFs showed the highest proliferative activity on day 7. However, the proliferative activity of each type of stromal cells was not significantly different (Figure 1b).

### 2.2. 2D Co-Culture of HSC-2 Cells and Various Types of Stromal Cells

#### 2.2.1. Effect of Stromal Conditioned Medium (CM) on the Growth of HSC-2 Cells

To investigate the effect of tumor stroma on HSC-2 proliferation, the growth ability of HSC-2 was measured added to the stromal CM. On culture days 3 and 5, no difference was seen in cell proliferation between the control group and the groups to which HDF CM, ED-st CM, and EX-st CM was added. On culture day 7, the cell proliferation of all samples to which stromal CM had been added was higher than that of the control group (Figure 2a).

#### 2.2.2. Ki-67 Labeling Index

To investigate the proliferative activity of cancer cells when the cancer cells were co-cultured with various types of stromal cells, stromal cells and HSC-2 cells were co-cultured on a glass slide, and proliferation of the cancer cells was compared using the Ki-67 labeling index. The proliferative activity was in the order of HSC-2/ED-st, HSC-2/EX-st, and HSC-2/HDF, and the proliferative activity was significantly higher in the HSC-2/ED-st group than in the other groups (Figure 2b).

#### 2.2.3. Stromal Cells Enhance the Invasive Ability of Cancer Cells

To investigate the invasion activity of cancer cells when the cancer cells were co-cultured with various types of stromal cells, we compared the invasive ability between the HSC-2 alone group and various HSC-2/stroma co-cultured groups. The invasive ability was the highest in the HSC-2/ED-st co-cultured group, followed by the HSC-2/EX-st group, HSC-2/HDF group, and HSC-2 alone group. The total number of invading cells including tumor cells and stromal cells in the HSC-2/ED-st group was significantly higher than in the other groups (Figure 3a). However, because this cell count included both tumor cells and stromal cells, we used fluorescent double staining to differentiate between tumor cells and stromal cells, and we counted only the infiltrating tumor cells. The number of infiltrating cells was significantly higher in the HSC-2/ED-st co-cultured group than in the other co-cultured groups (Figure 3b).

#### 2.2.4. Comparison of the Morphology in 2D Co-Culture of HSC-2 Cells and Various Types of Stromal Cells

In the HSC-2/HDF co-culture group, cancer cells that formed a very small clusters were observed, the density of stromal cells was loose, and the stroma proliferative activity was low. In the HSC-2/EX-st co-culture group, large cancer nests were observed, and stroma growth was observed. In the HSC-2/ED-st co-culture group, relatively small cancer nests were formed, and vigorous growth of the cancer stroma was observed (Figure 4a).

The morphology and distribution of both tumor cells and stromal cells were examined in more detail following fluorescent double staining. In the HSC-2/HDF co-culture group, a small number of small tumor nests with weak adhesion between tumor cells was observed, and the growth of the tumor stroma was also loose. In the HSC-2/EX-st group, tumor cells formed large tumor nests, and the stromal cells were dendritic or spindle shaped. The stroma showed dense growth in some areas. In the HSC-2/ED-st group, vigorous growth of small tumor nests was observed, and dense dendritic tumor stromal cells were observed (Figure 4b).

### 2.3. Tumor Xenograft Model

#### 2.3.1. Histological Findings

Figure 5a shows representative histological findings in the tumor-implanted mouse model. In the HSC-2 alone group, tumor tissue formation with a large parakeratinized tumor nest was observed. The histological findings of the HSC-2/HDF group showed well-differentiated squamous cell carcinoma with islands of tumor cells, clearly visible squamous differentiation, and mild nuclear and cellular pleomorphism. In the HSC-2/EX-st group, the forming tumor tissue showed verrucous squamous cell carcinoma-like morphology with dendritic stroma. The tumor tissue showed marked surface keratinization, so-called church-spire keratosis, which is a hallmark of verrucous squamous cell carcinoma. The HSC-2/ED-st group showed the histological findings of well to moderately differentiated squamous cell carcinoma with islands and cords of tumor cells, evident squamous differentiation, and moderate nuclear and cellular pleomorphism.

#### 2.3.2. Differences in the Ki-67-Positive Rate among Different Types of Stroma

The Ki-67 labeling index in both the HSC-2/ED-st and HSC-2/EX-st groups was significantly higher than that of the HSC-2 alone group and the HSC-2/HDF group (Figure 5b).

#### 2.3.3. Effect of Different Stroma Properties on Bone Resorption

To investigate the effect of stromal cells on HSC-2 bone resorption ability, we compared the bone resorption ability between various HSC-2/stroma co-cultured groups in vivo using a tartrate-resistant acid phosphatase (TRAP) stain. Figure 6a shows representative microscopic images of invasive bone destruction observed in the tumor-implanted mouse model. In the HSC-2/HDF group, the border line between the tumor and the cranial bone was clear, and bone resorption was not observed. On the other hand, in the HSC-2/EX-st group, slight bone destruction and severe bone destruction were observed. In the HSC-2/ED-st group, cranial bone resorption was remarkable, and the tumor invasive ability was high.

To investigate bone destruction by osteoclasts, TRAP staining was carried out in the tumor-implanted mouse model. The number of multinucleated giant cells in the bone resorption area was higher in the HSC-2/ED-st group than in the HSC-2/EX-st group. In the HSC-2/HDF group, bone resorption was hardly observed (Figure 6b).

## 3. Discussion

The results of the present study demonstrate that the tumor stroma is directly associated with changes in the biological characteristics of the tumor parenchyma such as proliferation, invasion, and morphology, both in vitro and in vivo.

The tumor stroma has a profound impact on the development and progression of malignant tumors. However, the specific mechanisms associated with its activation and effects on regulation of tumorigenesis remain unclear [21]. Previous studies showed that the tumor parenchyma alters the characteristics of the tumor stroma in various cancers [22,23,24]; this has also been shown in OSCC [25]. The neoplastic changes that occur in the epithelium are followed by changes in the stroma surrounding the tumor cells, and the stroma promotes the differentiation of fibroblasts into myofibroblasts [26,27,28]. However, no studies have reported that the tumor stroma changes the biological characteristics of tumor cells. Our co-culture model in vitro and xenograft model in vivo are well suited for investigating influences on tumor parenchyma characteristics by the stroma.

### 3.1. Properties of the Tumor Stroma Isolated from Different Subtypes of OSCCs

The stromal cells isolated from two types of human OSCC with different invasiveness were positive for vimentin. These stromal cells did not contain epithelial components because these stroma cells were not positive for AE1/3 (Figure 1a). Both ED-st and EX-st showed a similar spindle-shaped morphology, but both stromal cell types were partially α-SMA positive. Thus, the established tumor stroma was considered to be a heterogeneous cell population, and our experimental system reproduced the actual state of the tumor stroma.

Some researchers have reported that α-SMA is widely expressed in CAFs or myofibroblasts and that expression of α-SMA is associated with poor prognosis of patients with various types of cancers, including OSCC [29,30,31,32,33]. As the key component of the tumor stroma, CAFs are important regulators of tumor growth, angiogenesis, invasion, metastasis, and poor prognosis in many malignancies [34,35,36,37,38,39]. CAFs alone established from human tissue were used in a previous study that examined the interaction between the tumor parenchyma and stroma. However, the actual tumor stroma consists of heterogeneous populations of cells. This experiment was different from an experiment using stroma containing only CAFs as in previous studies [26,40,41].

The tumor stroma is educated by the tumor parenchyma. CAFs and myofibroblasts are abundant in invasive OSCC; on the other hand, small amount of these cells are detected in oral verrucous carcinoma, which is consistent with the hypothesis that oral verrucous carcinoma is a form of well-differentiated squamous cell carcinoma with specific clinical and histological features, including slow growth, no invasive potential, and minimal tendency to metastasize [42]. In this study, we compared the proliferative ability of different types of tumor stroma and found no significant difference (Figure 1b). However, the α-SMA-positive rate was significantly higher in ED-st than in EX-st (Figure 1a), suggesting that the stroma we isolated reflects clinically relevant aspects of the OSCC stroma. In our previous study comparing EX-st and ED-st in human tissues, the α-SMA-positive rate was significantly higher in ED-st than in EX-st, consistent with the results of this study [43].

### 3.2. Effect of the Tumor Stroma on the Proliferative and Invasive Abilities of the Tumor Parenchyma

Next, to investigate the effect of the tumor stroma on the biological characteristics of the tumor parenchyma, especially the invasive and proliferative abilities of the tumor parenchyma, we performed experiments both in vitro and in vivo. Our in vitro experiments showed that stromal CM increased the proliferation of tumor cells at day 7 (Figure 2a). In particular, EX-st CM increased tumor cell proliferation in a concentration-dependent manner at day 7 (Figure 2a). The invasive ability of tumor cells was promoted when tumor cells were co-cultured with stromal cells (Figure 3).

In the xenograft model, the tumor stroma also remarkably increased the proliferative activity of tumor cells as measured with the Ki-67 labeling index (Figure 5b), and the invasive ability of tumor cells in bone resorption as seen with TRAP staining (Figure 6b).

A comparison of the HSC-2/ED-st co-culture group and HSC-2/EX-st co-culture group in vitro and in vivo showed that ED-st more strongly promoted tumor cell growth and more dramatically stimulated the invasive ability of tumor cells than EX-st. This effect was not detected when HDFs were co-cultured with HSC-2 cells in the same experimental conditions (Figure 2 and Figure 3). From these data, we conclude that in this co-culture system, tumor stroma stimulated progression of OSCC. In addition, the present experimental results indicated that ED-st, which was isolated from invasive OSCC, promoted the growth and invasive ability of HSC-2 cells to a greater extent than EX-st, which was isolated from verrucous OSCC because the same OSCC cell line, HSC-2, was used in these experiments. Therefore, the tumor stroma, which was educated by the tumor parenchyma in the human tumor microenvironment, retained this original property and continued to maintain its characteristics, even at the transplant destination, and influenced the biological characteristics of the tumor parenchyma.

Several recent studies have shown that tumor cells transplanted with the stroma in some tumors, such as pancreatic tumors, alter the ability of the tumor to spread and influence its prognosis [30,31,32,33,34,35,36,37,38,39,44]. In this regard, these studies have shown results similar to our experimental results. However, previous studies have stated that the tumor stroma affects tumor cell invasion and proliferation, but no studies have shown that the tumor stroma alters the histological structure or morphology.

### 3.3. Effect of the Tumor Stroma on the Morphology of the Tumor Parenchyma

To further investigate the effect of the tumor stroma on the biological characteristics of tumor parenchymal morphology or the tumor tissue structure, we performed experiments both in vivo and in vitro. In the HSC-2/HDF co-culture group, a small number of small tumor nests with weak adhesion between tumor cells was observed. In the HSC-s/EX-st group, tumor cells formed large tumor nests and, in the HSC-s/ED-st group, vigorous growth of small tumor nests was observed (Figure 4b). In the xenograft model in vivo, the tumor stroma dramatically altered the histological structure of the tumor tissue. The histological findings of the HSC-2/HDF group showed well-differentiated squamous cell carcinoma with islands of tumor cells, clearly visible squamous differentiation, and mild nuclear and cellular pleomorphism. In the HSC-2/EX-st group, the forming tumor tissue showed verrucous squamous cell carcinoma-like morphology with dendritic stroma. The tumor tissue showed marked surface keratinization, so-called church-spire keratosis, which is a hallmark of verrucous squamous cell carcinoma. On the other hand, the histological findings of the HSC-2/ED-st groups showed well to moderately differentiated squamous cell carcinoma with islands and cords of tumor cells, evident squamous differentiation, and moderate nuclear and cellular pleomorphism (Figure 5a).

These results indicated that the tumor stroma transmitted the morphology of the tumor tissue in which it was originally present to the transplanted tissue. Therefore, these results suggested that the tumor stroma has the potential to directly control the biological morphology of the tumor parenchyma. The results of this experiment differ from the conventional concept in oncology that the tumor parenchyma controls the tumor stroma, suggesting the development of new treatment strategies. However, there may be some possible limitations in this study. The issue that stromal cells were derived from only two patients for this study, which bring the potential for issues concerning reproducibility across patient samples. In order to improve the reproducibility of this experiment, it is necessary to further increase the number of samples and generalize the experimental results.

Taken together, the present study is the first to demonstrate that the tumor stroma can enhance various malignant properties of an OSCC cell line in vitro and in vivo. Moreover, regarding previously published data on the interaction between the tumor parenchyma and tumor stroma, our results add more insight into the potential oncological mechanism.

## 4. Materials and Methods

In this study, experiments were organized into two parallel series. In the first series, the influence of the stroma was examined with analyses of morphology, proliferation, and invasive activity using an in vitro co-culture system. In the second series, the same influence was examined in a xenograft model in vivo.

### 4.1. OSCC Cell Line

HSC-2 cells were purchased from the JCRB Cell Bank (Osaka, Japan) and maintained in Dulbecco’s Modified Eagle’s Medium (DMEM) (Life Technologies, Thermo Fisher Scientific Inc., KK, Tokyo, Japan) supplemented with 10% fetal bovine serum (FBS), and 100 U/mL antimycotic–antibiotic (Life Technologies, Thermo Fisher Scientific Inc., KK, Tokyo, Japan) at 37 °C in a humidified atmosphere with 5% CO_2_.

### 4.2. Stroma Isolation

The method of isolation of stromal cells is illustrated in Figure 7. Preparation of primary cultured stroma from human OSCC tissues was performed. The stromal cells samples were obtained from surgical operative tissues in the Oral Surgery Department of Okayama University. This study was approved by the Ethics Committee of Okayama University (the project identification code: 1703-042-001). In addition, we obtained informed consent from all patients. Pieces of fresh oral squamous carcinoma tissue (1 mm^3^) were washed several times with Alpha-MEM (Life Technologies, Thermo Fisher Scientific Inc.) containing antibiotic–antimycotic (Life Technologies, Thermo Fisher Scientific Inc.) and then minced. These tissues were treated with Alpha-MEM containing 1 mg/mL collagenase II (Invitrogen Co., New York, NY, USA) and Dispase (Invitrogen Co., New York, NY, USA) for 2 h at 37 °C with shaking (200 rpm). The released cells were centrifuged for 5 min at 1000 rpm, suspended in Alpha-MEM containing 10% FBS (Biowest, Nuaillé, France), filtered using a Cell strainer (100 μm, BD Falcon: BD Bioscience, Primus, UK), plated in a tissue culture flask, and incubated at 37 °C in 5% CO_2_. One week later, stromal cells were obtained following treatment with Accutase (Invitrogen Co.) based on the different adhesion of epithelial and stromal cells. We named these stromal cells ED-st (derived from ED-type OSCC stroma) and EX-st (derived from EX-type OSCC stroma). In this experiment, human dermal fibroblasts (HDFs) purchased from LONZA (Tokyo, Japan) were used as a control for the tumor stroma. These stromal cells and HDFs were maintained in Alpha-MEM containing 10% FBS and were used within 10 passages to eliminate transformation due to passaging.

### 4.3. In Vitro Experiment

#### 4.3.1. Tumor Stromal Properties

##### Tumor Stromal Morphology

Various stromal cells were cultured in 60-mm dishes for 72 h. Giemsa staining was carried out with a commercial kit (Diff-Quick, Nanjing Jiancheng Bioengineering Institute, Nanjing, China). Stained cells were observed and photographed with a bright field microscope (×100 magnification) (BX51, Olympus, Tokyo, Japan).

##### Tumor Stroma Proliferation Assay (MTS Assay)

Various stromal cells (2.0 × 10^3^ cells/100 μm/well) were cultured in 96-well plates (CELLSTAR, Greiner Bio-One, Frickenhausen, Germany) in Alpha-MEM. After the cells had adhered to the plates, stromal cells were cultured in Alpha-MEM containing 10% FBS for 1, 3, or 7 days in a 5% CO_2_ atmosphere at 37 °C. Cell proliferation was assessed using the MTS assay according to the manufacturer’s recommendations (CellTiter 96 Aqueous One Solution Cell Proliferation Assay, Promega Corporation, Madison, WI, USA).

#### 4.3.2. Effect of Stromal Conditioned Medium (CM) on HSC-2 Cell Proliferation

HSC-2 cells (2.0 × 10^3^ cells/100 μm/well) were seeded in 96-well plates (CELLSTAR, Greiner Bio-One) in Alpha-MEM, and stromal cell CM of different concentrations (0, 10, 20, 30, 40, 50%) was added. After the cells had adhered to the plates, HSC-2 cells were cultured for 1, 3, or 7 days in a 5% CO_2_ atmosphere at 37 °C. Cell proliferation was assessed using the MTS assay according to the manufacturer’s recommendations (CellTiter 96 Aqueous One Solution Cell Proliferation Assay, Promega Corporation).

### 4.4. Co-Culture In Vitro

#### 4.4.1. Cell Morphology

Various stromal cells (3.0 × 10^5^ cells/5 mL) were seeded in a 60-mm dish, and HSC-2 cells (1.0 × 10^5^ cells/5 mL) were added. Morphological changes of the HSC-2 cells were assessed at 3 days with Giemsa staining and fluorescent double staining.

#### 4.4.2. Double-Fluorescent Immunohistochemical Staining

Various stromal cells were cultured in 60-mm dishes with glass coverslips for 72 h. Tumor stroma and HDFs were fixed with 4% paraformaldehyde and incubated with mouse anti-α-smooth muscle actin (SMA) (ACTA2) (1:200, Abcam, Cambridgeshire, UK), rabbit anti-vimentin (SP20) (1:200, Abcam), and mouse anti-pan Cytokeratin (AE1/3) (Abcam) overnight at 4 °C. Tumor stroma was then incubated with anti-mouse IgG Alexa Fluor 488 (1:100, Life Technologies, Carlsbad, CA, USA) for AE1/3 and α-SMA or anti-rabbit IgG Alexa Fluor 568 (1:100, Life Technologies) for vimentin at room temperature for 1 h. After the reactions, the sections were stained with 0.2 g/mL 4′,6-diamidino-2-phenylindole (DAPI) (Dojindo Laboratories, Kumamoto, Japan). The staining results were observed with an All in One, BZ ×700 fluorescence microscope (Keyence, Osaka, Japan).

#### 4.4.3. Invasion Assay

Cell invasion assays were carried out using 8-μm transwell filters that were precoated with Matrigel in 24-well plates (Corning BioCoat Matrigel Invasion Chamber kit, BD Biosciences, San Jose, CA, USA). Cells (HSC-2: 2.5 × 10^4^, stromal cells: 7.5 × 10^4^) were resuspended in 200 μL serum-free medium and added to the upper chamber, whereas the lower chamber was filled with Alpha-MEM including 10% FBS. After incubation for 24 h, the upper chambers were fixed with 4% paraformaldehyde, and the membranes were removed. The membranes were stained by Giemsa staining and double-fluorescent immunohistochemical staining with rabbit anti-vimentin (1:200, Abcam), and mouse anti-AE1/3 (Abcam). The membrane was removed and placed on a slide for microscopic observation. Migrated cells on the lower surface of the chamber were counted and photographed (BX51, Olympus, Tokyo, Japan). Cells were then counted under an inverted light microscope in three independent fields at ×100 magnification. Experiments were performed four times independently.

### 4.5. In Vivo Experiment (Tumor Xenografts)

#### 4.5.1. Experimental Animals

All animal experiments were performed in accordance with relevant guidelines and regulations and were approved by the institutional committees at Okayama University (OKU-2017406). A total of 14 BALB/c nude mice (4-week-old healthy females) were used in this experiment. BALB-c nu-nu female mice were injected subcutaneously into the head with 1.0 × 10^6^ HSC-2 cells and 3.0 × 10^6^ stromal cells (HDFs, ED-st, or EX-st). For implantation of cells, mice were anesthetized intraperitoneally with ketamine hydrochloride (75 mg/kg body weight) and medetomidine hydrochloride (0.5 mg/kg body weight), and we confirmed that mice were anesthetized. Atipamezole hydrochloride (1 mg/kg body weight) was injected subcutaneously to induce awakening.

#### 4.5.2. Histological Examination

For histological observations, implanted tumor tissue was removed after 4 weeks, and all embedded tissues were fixed in 4% paraformaldehyde for 12 h. The specimens were decalcified in 10% EDTA for 3 weeks. Tissues were processed and embedded in paraffin wax via routine histological preparation and sectioned at 5-μm thickness. The sections were used for hematoxylin–eosin staining.

#### 4.5.3. Cell Proliferation Assay (Ki-67 Labeling Index)

For in vivo tissue xenografts, proliferation measurements in paraffin-embedded samples of the tumor explants were performed using Ki-67, a proliferation-specific marker. Endogenous peroxidase activity was blocked by incubating the sections in 0.3% H_2_O_2_ in methanol for 30 min. Antigen retrieval was achieved by heat treatment using 10 mM citrate buffer solution (pH 6.0). After treatment with horse normal serum, the sections were incubated with primary antibodies at 4 °C overnight. Mouse monoclonal anti-Ki-67 (Mib1) (1:50, Abcam) and rabbit anti-vimentin (1:50, Dako, Carpinteria, CA, USA) were used as primary antibodies. The secondary antibody was applied for 1 h at room temperature. Immunoreactivity was visualized using diaminobenzidine (DAB)/H_2_O_2_ solution (Histofine DAB substrate; Nichirei, Tokyo, Japan), and the section was counterstained with Mayer’s hematoxylin. Ki-67 expression was evaluated by calculating the labeling index (the number of positive cells/total cells × 100).

#### 4.5.4. Tartrate-Resistant Acid Phosphatase (TRAP) Staining

Staining for TRAP was carried out using the TRAP staining kit (Primary Cell, Sapporo, Japan) according to the manufacturer’s instructions.

### 4.6. Statistical Analysis

Statistical analysis was performed using one-way analysis of variance and Fisher’s exact tests. A *p* value < 0.05 was considered statistically significant. All calculations were performed using PASW Statistics 18 (SPSS Inc., Chicago, IL, USA).

## 5. Conclusions

In the present study, we isolated human primary stroma from patients with OSCC and discovered that the stroma promoted proliferation and invasion of HSC-2 cells in vitro and changed the biological characteristics of the tumor in vivo. Furthermore, the origin of the stroma was strongly associated with tumor growth and the biological characteristics of the tumor.

## Figures and Tables

**Figure 1 ijms-21-07714-f001:**
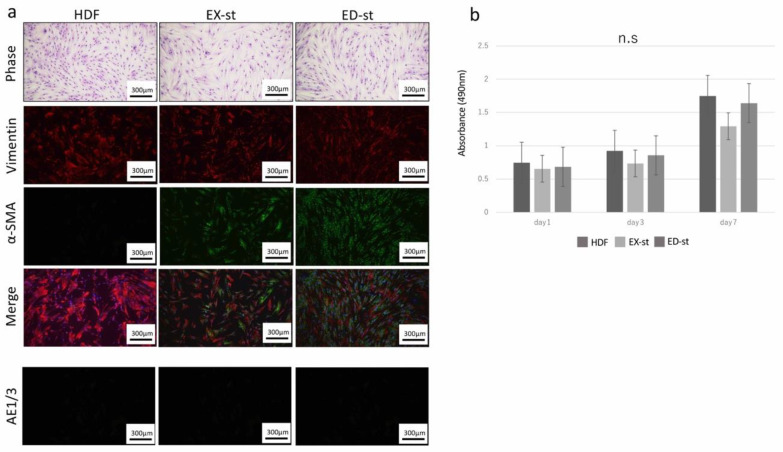
Generation of tumor stroma from human oral squamous cell carcinoma (OSCC). (**a**) Isolated tumor stroma had irregular spindle-shaped morphology, whereas dermoid fibroblasts (HDF)s had regular short spindle-shaped morphology. Representative images of immunofluorescence staining of cultured stroma and HDFs for vimentin (red) and α-smooth muscle actin (SMA) (green) and DAPI counterstaining (blue), and mouse anti-pan Cytokeratin (AE1/3) (green). (**b**) No significant difference was found in the proliferative ability among the three types of cells on day 1, day 3, and day 7. n.s.; no significant difference. *n* = 8.

**Figure 2 ijms-21-07714-f002:**
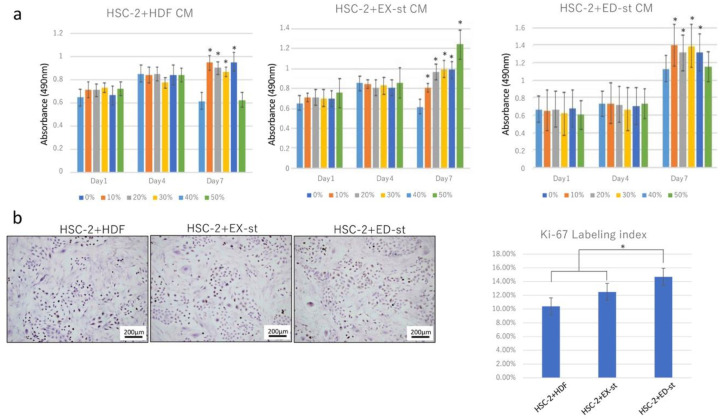
The tumor stroma promotes HSC-2 cell proliferation in vitro. (**a**) The growth of HSC-2 cells cultured with stromal conditioned medium was greatly accelerated compared with cells cultured with HDF conditioned medium, especially on day 7. Endophytic type OSCC (ED-st) conditioned medium induced a greater increase in proliferation of HSC-2 cells than exophytic type OSCC (EX-st) conditioned medium. *n* = 8. (**b**) The Ki-67 labeling index was determined in co-culture. The ED-st/HSC-2 co-culture group showed a higher Ki-67 labeling index than both the EX-st/HSC-2 and HDF/HSC-2 groups. *n* = 4, * *p* < 0.05.

**Figure 3 ijms-21-07714-f003:**
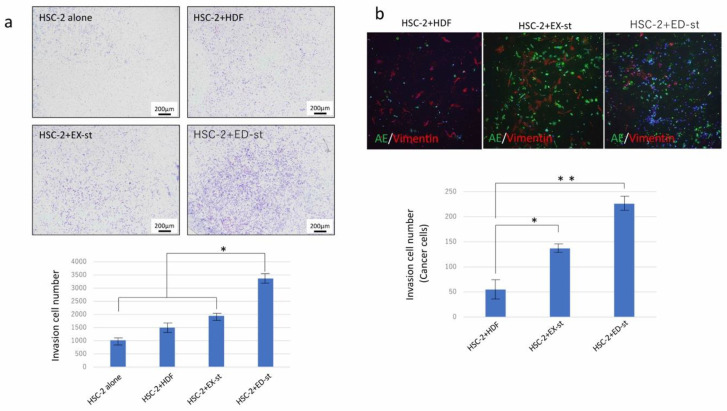
The tumor stroma promoted the invasive ability of HSC-2 cells in vitro. (**a**) Representative light microscopic images showing stained tumor stroma/HDFs and HSC-2 cells on the lower surface of the microporous transwell insert after invasion. The invasive ability of HSC-2 cells was strongly promoted when co-cultured with stromal cells compared to HDFs. ED-st in particular most strongly promoted the invasive ability of HSC-2 cells. *n* = 4. (**b**) Representative images of immunofluorescence staining of invading cells with vimentin (red) and α-SMA (green) staining and DAPI counterstaining (blue). The tumor stroma promoted the invasive ability of HSC-2 cells compared to HDFs, and ED-st most strongly increased the invasive ability of HSC-2 cells. *n* = 4, * *p* < 0.05, ** *p* < 0.01.

**Figure 4 ijms-21-07714-f004:**
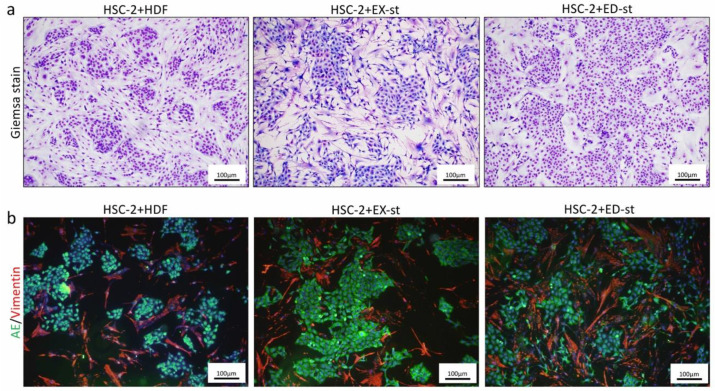
Representative images and immunofluorescent staining of 2D co-culture with HSC-2 cells and various types of stroma. (**a**) In the HSC-2/HDF co-culture group, cancer cells forming very small masses were observed. In the HSC-2/EX-st co-culture group, large cancer nests were observed. In the HSC-2/ED-st co-culture group, relatively small cancer nests were formed. (**b**) Immunofluorescent staining with vimentin (red) and AE1/3 (green) and DAPI counterstaining (blue). In the HSC-2/HDF co-culture group, a small number of small tumor nests with weak adhesion between tumor cells was observed, and the growth of the tumor stroma was also loose. In the HSC-s/EX-st group, tumor cells formed large tumor nests, the stromal cells showed dendritic or spindle-shaped morphology, and the stroma showed dense growth in some areas. In the HSC-s/ED-st group, vigorous growth of small tumor nests was observed, and dense dendritic tumor stromal cells were observed.

**Figure 5 ijms-21-07714-f005:**
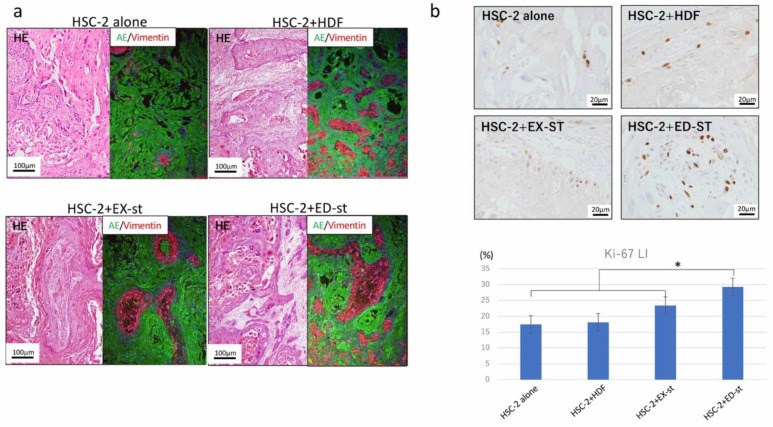
Histological images of the xenograft model and Ki-67 labeling index. (**a**) In the HSC-2 alone group, tumor tissue formation with marked keratinized tumor nests was observed. The histological findings of the HSC-2/HDF group showed islands of well-differentiated squamous cell carcinoma. In the HSC-2/EX-st group, the forming tumor tissue showed verrucous squamous cell carcinoma-like morphology with dendritic stroma. The tumor tissue showed marked surface keratinization, so-called church-spire keratosis. The HSC-2/ED-st group showed histological findings of well to moderately differentiated squamous cell carcinoma with islands and cords of tumor cells with evident squamous differentiation. (**b**) The Ki-67 labeling index in both the HSC-2/ED-st groups was significantly higher than that of the HSC-2 alone, the HSC-2/HDF and HSC-2/EX-st group. *n* = 4, * *p* < 0.05.

**Figure 6 ijms-21-07714-f006:**
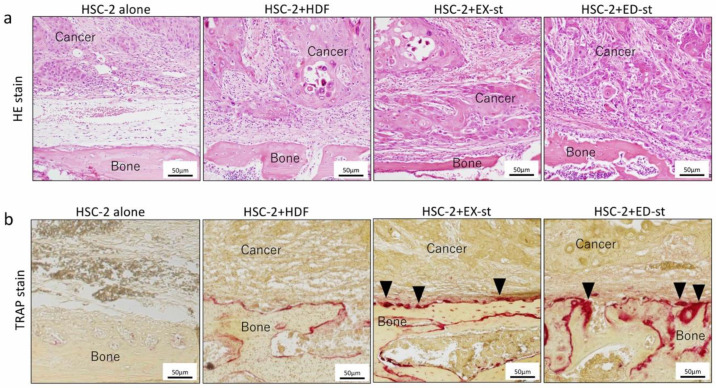
Histological images and tartrate-resistant acid phosphatase (TRAP) staining of tumor tissue at the bone surface in the xenograft model. (**a**) In the HSC-2/HDF group, the border line between the tumor and the cranial bone was clear, and bone resorption was not observed. In the HSC-2/EX-st group, slight bone destruction and severe bone destruction were observed. In the HSC-2/ED-st group, cranial bone resorption was remarkable, and the tumor invasive ability was high. (**b**) The number of multinucleated giant cells in the bone resorption area was higher in the HSC-2/ED-st group than in the HSC-2/EX-st group, as indicated by the arrowheads. In the HSC-2/HDF group, bone resorption was hardly observed.

**Figure 7 ijms-21-07714-f007:**
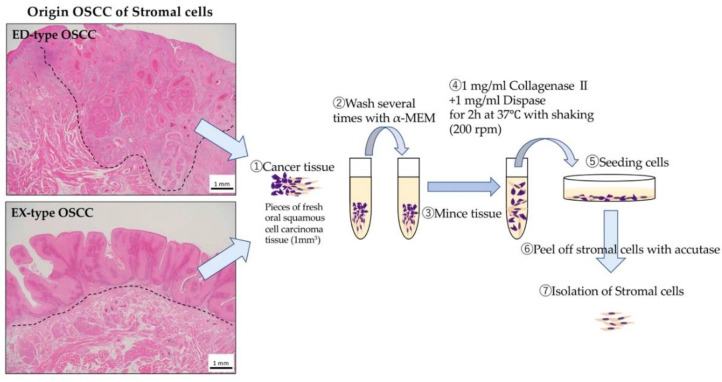
Isolation of various stromal cells. Pieces of fresh oral squamous carcinoma tissue (1 mm^3^) were washed several times with Alpha-MEM and then minced. These tissues were treated with Alpha-MEM containing 1 mg/mL collagenase II and Dispase for 2 h at 37 °C with shaking (200 rpm). The released cells were centrifuged for 5 min at 1000 rpm, suspended in Alpha-MEM containing 10% FBS, filtered using a Cell strainer, plated in a tissue culture flask, and incubated at 37 °C in 5% CO_2_. One week later, stromal cells were obtained following treatment with Accutase based on the different adhesion of epithelial and stromal cells.

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
