# Peer review of "Impact of the Stroma on the Biological Characteristics of the Parenchyma in Oral Squamous Cell Carcinoma"

_ijms, 2020, doi:10.3390/ijms21207714_

Round 1

Reviewer 1 Report

The present studies described the interesting phenomenon on how the tumor and the tumor stroma interact with each other. Often, the effect of tumor cells on stromal cells are described, while in this study the effect of stromal cells is described on tumor cells. Stromal cells were isolated from two types of oral squamous cell carcinoma (OSCC). Each type had a different growth pattern, with one growing more outward (EX-type) and the other one more invasive (ED-type). The effects on HSC-2 cells (OSCC cell line) when placed with stromal cells (or conditioned medium), from different OSCC types, were studied. Interestingly, stromal cells from invasive type of OSCC made HSC-2 cells proliferate more and made them more invasive compared to the stromal cells of the other OSCC type. This study showcases the direct influence that stromal cells have on tumor cells.

While going through the manuscript I had some major and minor comments which I have listed below.

Major comments:

  • It is not clear to me from how many different tumor samples these stromal cells are originating. I see that these cells are used for experiments until the 10th passage. Are replicated experiments performed with stromal cells from different patient in each replicate? Is the isolation of stromal cells from these patients always successful? Are tumor cells not contaminating your cultures?
  • A specific section on informed consent from patients and how the material was obtained is missing. Where these biopsies taken especially for this experimental purpose? Or how much of the biopsy was obtained?
  • Figure 2a, 3a-b, and other figures: How many replicates were done for this experiment? Please show the error bars.
  • Figure 2: Interesting result that so little tumor cells are present in b. in figure a I see 15-20x more cells. What ratio was used for seeding the cells in upper chamber? The stromal cells seem to be migrating much more.
  • Line 218 and 235: You can only state this when you also stain the original biopsy with vimentin and a-SMA. Please do so.
  • Line 234: As this was not quantified in the manscript/figure, how can it be significant?
  • Line 300: what was the orginal growth pattern of the tumor from where the HSC-2 cell line is established? EX or ED? would have been nice to test cells that originally had the opposite growth type to see what the stromal cells do to this type of cancer cell. 

Minor comments:

  • Line 44-46: questionable if references from 2010-2014 are to be called "recent"
  • Line 55: include references for the following statements about EX- and ED-types.
  • Line 61: “macroscopic subtypes are defined by cancer parenchyma properties”. Please give an example of these properties and include references.
  • Line 64-67: is the recurrence not related to the previous tumor that was removed when looking at the mutational profile? Is the tumor stroma also not removed when removing the tumor + healthy epithelium around it? Does the recurrence not just occur by remaining tumor cells that were not completely removed by the surgery? I also miss references for this part. Are you stating that leftover tumor stroma regenerates a new tumor?
  • Line 77 and rest of manuscript: use of “stroma cells” or “stromal cells”
  • Line 78: abbreviations for EX-st and ED-st are not explained yet (only in method sections but this come at the end)
  • Line 81-82: it would be nice to show this data in supplemental next to a panCK and CD45 stain. Maybe also add FAP? in addition it would be nice to show the expression of these markers in the original tumor biopsy where these cells were coming from.
  • Line 86: numbering issue --> "2.2 2-D" I assume
  • Section 2.2.1: explanation and rationale of the experiment here is missing. I assume HSC-2 cells were co-cultured with conditioned medium of stromal cells
  • Line 88: I don't see results for day 3 and 5, only 1, 4 and 7. For clarity maybe leave out day 4 as well and mention that there are no significant differences
  • Figure 1b, 2a, -c, 5b : no title for Y-axis.
  • Line 114: in M&M you describe that you also stained transwell membrane, where are the results for this?
  • Line 134: I wouldn't call this a very small mass à relatively small clusters
  • Figure 5a: Also perform double IF stain with vimentin and panCK to better compare with fig4b.
  • Line 181: Mention number of mice used per group in legend. And throughout the manuscript it should be mentioned in the figure legends at how many replicates we are looking at.
  • Figure 6: How did this look for HSC-2 cells alone? This would be good to compare with as tumor experiments with mice are usually carried out with tumor cells only.
  • Section 2.3.3: Mention rationale for the experiment first.
  • Line 216: Show in supplemental data together with data from my comment about line 81-82.
  • Line 229: If these cells are not detected in verrucous carcinoma, why do you see VIM+ and SMA+ cells in the biopsy of this type of tumor?
  • Line 240: Concentration dependant only holds true for EX-st not for ED-st.
  • Line 242: fig 2b doesn't say anything about invasiveness, but is about proliferation.
  • Line 306: and throughout manuscript: superscripts and subscripts are lost (maybe only this version though).
  • Line 323: does the commercial kit have a name?
  • Line 324: mention brand of microscope and type.
  • Line 345: were the cells grown on glass coverslips?
  • Line 345 and line 384: mention clone names for all antibodies. And also dilution for Ki-67
  • Line 346: the clone name is AE1/3 the actual protein it targets is pan-cytokeratin (panCK)
  • Line 347: mention incubation times and temperatures.
  • Line 353: transwell brand/company? in what plate? 96/24 well?
  • Line 354: what kind of cells and what ratio was used? 1:3 as in mice experiments?
  • Line 355: before it was called Alpha-MEM use one consistantly.
  • Line 357: next to Giemsa also IF was performed I believe.
  • Line 357-358: how were these cells stained?
  • Line 383: normal serum from what?

Author Response

The present studies described the interesting phenomenon on how the tumor and the tumor stroma interact with each other. Often, the effect of tumor cells on stromal cells are described, while in this study the effect of stromal cells is described on tumor cells. Stromal cells were isolated from two types of oral squamous cell carcinoma (OSCC). Each type had a different growth pattern, with one growing more outward (EX-type) and the other one more invasive (ED-type). The effects on HSC-2 cells (OSCC cell line) when placed with stromal cells (or conditioned medium), from different OSCC types, were studied. Interestingly, stromal cells from invasive type of OSCC made HSC-2 cells proliferate more and made them more invasive compared to the stromal cells of the other OSCC type. This study showcases the direct influence that stromal cells have on tumor cells.

While going through the manuscript I had some major and minor comments which I have listed below.

Major comments:

It is not clear to me from how many different tumor samples these stromal cells are originating. I see that these cells are used for experiments until the 10th passage. Are replicated experiments performed with stromal cells from different patient in each replicate? Is the isolation of stromal cells from these patients always successful? Are tumor cells not contaminating your cultures?

→We have added to the schema of the method of primary culture of stromal cells in Figure 7. The stromal cells used in this experiment were collected from one person each for both ED-st and EX-st. These stromal cells did not contain tumor cells because the stromal cells were AE1/3 negative in Fig 1a.

A specific section on informed consent from patients and how the material was obtained is missing. Where these biopsies taken especially for this experimental purpose? Or how much of the biopsy was obtained?

→The stroma cells samples were obtained from surgical operative tissues in the Oral Surgery Department of Okayama University. This study was approved by the Ethics Committee of Okayama University Graduate School of Medicine, Dentistry and Pharmaceutical Sciences (the project identification code: 1703-042-001; date of approval: 13 March 2017; and name of the ethics committee: Analysis of biological property of oral cancer). And we obtained informed consent from all patients. We have added to the text in Materials and Methods. Line371-374.

Figure 2a, 3a-b, and other figures: How many replicates were done for this experiment? Please show the error bars.

→We have added to the error bars in Fig. 2a, 3a-b. We conducted Fig. 2a experiment 8 times, and conducted Fig. 3a-b experiments 4 times. We have added to the number of replicate times in figure legends.

Figure 2: Interesting result that so little tumor cells are present in b. in figure a I see 15-20x more cells. What ratio was used for seeding the cells in upper chamber? The stromal cells seem to be migrating much more.

→We have modified and added to the Methods of invasion assay in Materials and Methods. Line 431.

Line 218 and 235: You can only state this when you also stain the original biopsy with vimentin and a-SMA. Please do so.

→In our previous study, we have already investigated and confirmed the population of EX-st and ED-st by staining vimentin, α-SMA and so on. We have modified the text and have added to the reference. Line 285-286.

Line 234: As this was not quantified in the manscript/figure, how can it be significant?

→Fig. 1a showed that almost all stromal cells were vimentin positive and some of the stromal cells were α-SMA positive. Therefore, we thought that the stromal cells contained at least two cell types.

Line 300: what was the orginal growth pattern of the tumor from where the HSC-2 cell line is established? EX or ED? would have been nice to test cells that originally had the opposite growth type to see what the stromal cells do to this type of cancer cell.

→HSC-2 cell line has neither invasive nor metastatic potential commercially. We agree with your comments. We will consider an experimental system using another growth pattern cell line in the future.  

Minor comments:

Line 44-46: questionable if references from 2010-2014 are to be called "recent"

→We have modified the text. Line 44.

Line 55: include references for the following statements about EX- and ED-types.

→We have added to the references. Reference number 14,15.

Line 61: “macroscopic subtypes are defined by cancer parenchyma properties”. Please give an example of these properties and include references.

→We have modified the text and have included references. Line 62-63.

Line 64-67: is the recurrence not related to the previous tumor that was removed when looking at the mutational profile? Is the tumor stroma also not removed when removing the tumor + healthy epithelium around it? Does the recurrence not just occur by remaining tumor cells that were not completely removed by the surgery? I also miss references for this part. Are you stating that leftover tumor stroma regenerates a new tumor?

→Some cases of squamous cell carcinoma arising in the center of free flaps have been reported by several authors. So, we hypothesized that the remaining tumor stroma was affecting the healthy epithelium. We have added to the references. Reference number 19,20.

Line 77 and rest of manuscript: use of “stroma cells” or “stromal cells”

→We have changed stromal cells.

Line 78: abbreviations for EX-st and ED-st are not explained yet (only in method sections but this come at the end)

→We have added to the abbreviations for EX-st and ED-st. Line 80-81.

Line 81-82: it would be nice to show this data in supplemental next to a panCK and CD45 stain. Maybe also add FAP? in addition it would be nice to show the expression of these markers in the original tumor biopsy where these cells were coming from.

→We agree with you and we will consider to do in the future.

Line 86: numbering issue --> "2.2 2-D" I assume

→We means 2D (2 dimention) co-culture.

Section 2.2.1: explanation and rationale of the experiment here is missing. I assume HSC-2 cells were co-cultured with conditioned medium of stromal cells

→We have added to the text. Line 91-92.

Line 88: I don't see results for day 3 and 5, only 1, 4 and 7. For clarity maybe leave out day 4 as well and mention that there are no significant differences

→We did not measure cell proliferation for day 3 and 5. We measured cell proliferation every 3days. There were no significant differences between day 1 and 4, however we thought that the cell proliferation result data should be issued over time.

Figure 1b, 2a, -c, 5b : no title for Y-axis.

→We have modified the figures.

Line 114: in M&M you describe that you also stained transwell membrane, where are the results for this?

→We have stained transwell membranes in Fig. 3 results (invarion assay).

Line 134: I wouldn't call this a very small mass à relatively small clusters

→We have modified the text. Line 151

Figure 5a: Also perform double IF stain with vimentin and panCK to better compare with fig4b.

→We have modified the figures.

Line 181: Mention number of mice used per group in legend. And throughout the manuscript it should be mentioned in the figure legends at how many replicates we are looking at.

→We have mentioned the number of mice in legend. And we have added to the replicates numbers in the figure legends. 

Figure 6: How did this look for HSC-2 cells alone? This would be good to compare with as tumor experiments with mice are usually carried out with tumor cells only.

→We have added to the HSC-2 alone in Fig. 6.

Section 2.3.3: Mention rationale for the experiment first.

→We have added to the text. Line 203-204.

Line 216: Show in supplemental data together with data from my comment about line 81-82.

→We have added to the data in Fig 1a.

Line 229: If these cells are not detected in verrucous carcinoma, why do you see VIM+ and SMA+ cells in the biopsy of this type of tumor?

→In our previous study, small amount of SMA positive stromal cells were detected in EX type OSCC like verrucous carcinoma. Therefore, there is no contradiction even if EX-st contain SMA-positive cells. We have corrected the text and added to the reference. Line 285-286.

Line 240: Concentration dependant only holds true for EX-st not for ED-st.

→We have corrected the text. Line 291-292.

Line 242: fig 2b doesn't say anything about invasiveness, but is about proliferation.

→We have corrected the text. Line 293.

Line 306: and throughout manuscript: superscripts and subscripts are lost (maybe only this version though).

→We have modified the characters throughout this manuscript.

Line 323: does the commercial kit have a name?

→We have added to the commercial kit name. Line 395.

Line 324: mention brand of microscope and type.

→We have added to the brand of microscope and type. Line 396-397.

Line 345: were the cells grown on glass coverslips?

→We have used glass coverslips. We have added to the text. Line 419

Line 345 and line 384: mention clone names for all antibodies. And also dilution for Ki-67

→We have added to the clone names and dilution for Ki-67.

Line 346: the clone name is AE1/3 the actual protein it targets is pan-cytokeratin (panCK)

→We have confirmed.

Line 347: mention incubation times and temperatures.

→We have added to the text. Line 424-425.

Line 353: transwell brand/company? in what plate? 96/24 well?

→We used the 24-well plates including the invasion assay kit. We have modified the text.Line 430.

Line 354: what kind of cells and what ratio was used? 1:3 as in mice experiments?

→HSC-2 and stromal cell were used 1:3 as in animal experiments. We have added to the text. Line 431.

Line 355: before it was called Alpha-MEM use one consistantly.

→We have changed Alpha-MEM.

Line 357: next to Giemsa also IF was performed I believe.

→We have corrected the text. Line 435-436.

Line 357-358: how were these cells stained?

→→We have added to the text. Line 436-437.

Line 383: normal serum from what?

→We used horse normal serum.

Reviewer 2 Report

This manuscript titled as “Impact of the Stroma on the Biological Characteristics of the Parenchyma in Oral Squamous Cell Carcinoma”, submitted by Takabatake, et al. studied the function of subtypes of stroma in cancer, which is an intriguing puzzle in other cancers (e.g. pancreatic cancer) too. This is a straightforward study to compare EX-st, and ED-st in tumor invasion, which will provide additional support to the current knowledge of fibroblasts subtypes.

  1. The same group has demonstrated the EX-st are SMA negative staining in human tissue. So are the SMA+ in cultured EX-st the artificial effect from culture?
  2. Evidence of no epithelial contamination in isolated EX-st, and ED-st should be demonstrated.
  3. For both figure 2 and figure 5, ki67 staining appear significantly low than growing cells in general. Any sign of cells going senescent?

Author Response

This manuscript titled as “Impact of the Stroma on the Biological Characteristics of the Parenchyma in Oral Squamous Cell Carcinoma”, submitted by Takabatake, et al. studied the function of subtypes of stroma in cancer, which is an intriguing puzzle in other cancers (e.g. pancreatic cancer) too. This is a straightforward study to compare EX-st, and ED-st in tumor invasion, which will provide additional support to the current knowledge of fibroblasts subtypes.

  1. The same group has demonstrated the EX-st are SMA negative staining in human tissue. So are the SMA+ in cultured EX-st the artificial effect from culture?

→In our previous study, small amount of SMA positive stromal cells were detected in EX type OSCC like verrucous carcinoma. Therefore, there is no contradiction even if EX-st contain SMA-positive cells. We added to the text and reference. Line 285-286.

  1. Evidence of no epithelial contamination in isolated EX-st, and ED-st should be demonstrated.

→We have added to the figure in Fig 1a.

  1. For both figure 2 and figure 5, ki67 staining appear significantly low than growing cells in general. Any sign of cells going senescent?

→Ki-67 was more than 10% in tumor cells and Ki-67 positive rate in this experiments was considered to be similar to in vivo, but not sign of senescent.

Reviewer 3 Report

The manuscript entitled “Impact of the Stroma on the Biological Characteristics of the Parenchyma in Oral Squamous Cell Carcinoma” which was contributed Takabatake et al. describes cellular interaction between oral squamous cell carcinoma and different stromal cells from commercial fibroblast cell line and patient derived fibroblast from endophytic (ED) type or exophytic (EX) type OSCC patients. The authors found ED-type fibroblast condition medium can promote oral cancer cell proliferation, invasion and other malignant behaviors in both in vitro and in vivo. It’s an interesting phenotype observation but unfortunately without deeper study to clarify the detail mechanism underline the study. Thus, the study the manuscript need to revised that can be accepted by IJMS.

Major comments:

  1. What kind of molecules or potential signaling proteins which help inter-cellular commutation has not reminding in the manuscript?
  2. In figure 2A, the cell growth rate in 0% condition medium should be the same. Because it only has cancer cell culture medium, HDF-CM and EX-st CM have the same pattern, why the ED-st CM can outgrowth in day 7? If the baseline is not consistency, it’s hard to say meaningful difference. Furthermore, there are no error bar in the figure. How many repeat in this experiments?
  3. Again, there are no error bar in figure 3A and B.
  4. The authors should remind AE1/3 role in the results section and stain AE1/3 in different type cells that can help reader understanding it’s function to distinguish cancer cells
  5. In figure4, why chemical staining by Giemsa stain reveal quite weak staining pattern in HSC-2 EX-st group but it has beautiful stain result by IF?
  6. In figure 5b, the counter stain is not clear in the photograph picture. Counter stain can help well-training reader understanding the cell type of proliferation cells.

Author Response

The manuscript entitled “Impact of the Stroma on the Biological Characteristics of the Parenchyma in Oral Squamous Cell Carcinoma” which was contributed Takabatake et al. describes cellular interaction between oral squamous cell carcinoma and different stromal cells from commercial fibroblast cell line and patient derived fibroblast from endophytic (ED) type or exophytic (EX) type OSCC patients. The authors found ED-type fibroblast condition medium can promote oral cancer cell proliferation, invasion and other malignant behaviors in both in vitro and in vivo. It’s an interesting phenotype observation but unfortunately without deeper study to clarify the detail mechanism underline the study. Thus, the study the manuscript need to revised that can be accepted by IJMS.

Major comments:

What kind of molecules or potential signaling proteins which help inter-cellular commutation has not reminding in the manuscript?

→You have raised an important point. We have attempted to compare EX-st and ED-st using microarray. We would like to write these results next manuscript.

In figure 2A, the cell growth rate in 0% condition medium should be the same. Because it only has cancer cell culture medium, HDF-CM and EX-st CM have the same pattern, why the ED-st CM can outgrowth in day 7? If the baseline is not consistency, it’s hard to say meaningful difference. Furthermore, there are no error bar in the figure. How many repeat in this experiments?

→We have added to the error bars in fig. 2a. In ED-st CM group, the cell growth rate in 0% CM was higher than EX-st CM and HDF CM group in 0%. That was because the error bar was a little bigger in  ED-st CM group than another groups. However, this experiment has been repeated 8 times, and I think the experimental results were highly reliable.

Again, there are no error bar in figure 3A and B.

→We have modified the fig 3a and b.

The authors should remind AE1/3 role in the results section and stain AE1/3 in different type cells that can help reader understanding it’s function to distinguish cancer cells

→We have added to the AE1/Vimentin double stain in fig 5a.

In figure4, why chemical staining by Giemsa stain reveal quite weak staining pattern in HSC-2 EX-st group but it has beautiful stain result by IF?

→We have changed the photo of HSC-2+EX-st in fig 4a.

In figure 5b, the counter stain is not clear in the photograph picture. Counter stain can help well-training reader understanding the cell type of proliferation cells.

→We have enlarged the photo of fig 5b.

Round 2

Reviewer 1 Report

Dear authors,

Thank you for taking my suggestions into consideration.

However, I feel like only minor revisions on the manuscript have been taking place, such as adding more information and textual adjustments.

My foremost concern is that scientific article can’t be based on only data derived from two patients. At least 3 per tumor type have to be included (6 in total) in order to be considered for publishing.

I looked up the characteristics of HSC-2 cell line and found that this cell line is derived from metastatic site: Cervical lymph node https://web.expasy.org/cellosaurus/CVCL_1287

However, you state that this cell line does not have invasive nor metastatic potential.

I wish you good luck with further future experiments.

Kind regards,

Author Response

Thank you for taking my suggestions into consideration.

However, I feel like only minor revisions on the manuscript have been taking place, such as adding more information and textual adjustments.

My foremost concern is that scientific article can’t be based on only data derived from two patients. At least 3 per tumor type have to be included (6 in total) in order to be considered for publishing.

→We agree that additional information on various samples as the reviewer suggested would be valuable. We are now investigating this point and intend to report it in a later paper. So we have added to the comment about limit of our manuscript in Discussion. Line 349-352.

I looked up the characteristics of HSC-2 cell line and found that this cell line is derived from metastatic site: Cervical lymph node https://web.expasy.org/cellosaurus/CVCL_1287

However, you state that this cell line does not have invasive nor metastatic potential.

→In our animal studies, HSC-2 did not have invasive ability (no bone resorption) and not metastatic potential.

Reviewer 3 Report

The authors have answered all my question. 
This manuscript can be accepted by IJMS.

Author Response

Thank you very much for providing important comments. We are thankful for the time and energy you expended.